# Exploiting the Fruitfly, *Drosophila melanogaster*, to Identify the Molecular Basis of Cryptochrome-Dependent Magnetosensitivity

Adam Bradlaugh [1,†] , Anna L. Munro [1,†] , Alex R. Jones [2] and Richard A. Baines [1,*]

1   Division of Neuroscience and Experimental Psychology, School of Biological Sciences, Faculty of Biology, Medicine and Health, University of Manchester, Manchester Academic Health Science Centre, Manchester M13 9PL, UK; adam.bradlaugh@manchester.ac.uk (A.B.); anna.munro@manchester.ac.uk (A.L.M.)
2   Biometrology, Department of Chemical and Biological Science, National Physical Laboratory, Teddington, Middlesex TW11 0LW, UK; alex.jones@npl.co.uk
*   Correspondence: Richard.Baines@manchester.ac.uk
†   Authors contributed equally.

**Abstract:** The flavoprotein CRYPTOCHROME (CRY) is now generally believed to be a magnetosensor, providing geomagnetic information via a quantum effect on a light-initiated radical pair reaction. Whilst there is considerable physical and behavioural data to support this view, the precise molecular basis of animal magnetosensitivity remains frustratingly unknown. A key reason for this is the difficulty in combining molecular and behavioural biological experiments with the sciences of magnetics and spin chemistry. In this review, we highlight work that has utilised the fruit fly, *Drosophila melanogaster*, which provides a highly tractable genetic model system that offers many advantages for the study of magnetosensitivity. Using this "living test-tube", significant progress has been made in elucidating the molecular basis of CRY-dependent magnetosensitivity.

**Keywords:** FAD; insect; magnetic field; animal magnetoreception; neuron; cryptochrome; *Drosophila*





## 1. Main Text

The precise biophysical origin of animal magnetoreception remains unclear. The radical pair mechanism (RPM) hypothesis of magnetoreception was first posited in the late 1970s, following the discovery that electron transfer and related processes can generate a pair of radicals with properties (singlet and triplet spin states) that can be affected by exposure to a magnetic field (MF) [1]. In 2000, Ritz first suggested that the blue-light (BL)-sensitive protein CRYPTOCHROME (CRY) might be the elusive magnetoreceptor in magnetically sensitive organisms [2]. This was based on the fact that the photochemistry of CRY is mediated by the photoexcitation of a bound cofactor, flavin adenine dinucleotide (FAD), and a subsequent electron transfer to FAD from a chain of neighbouring tryptophan residues, generating a radical pair (RP) consisting of a flavin semiquinone (FAD$^{\bullet-}$) and an oxidised Trp (TrpH$^{\bullet+}$) [3]. Electron transfer has been proposed to be mediated by a triad of Trp residues in CRY, although a fourth Trp residue has also recently been implicated, raising the idea of a Trp-tetrad and/or possible redundancy in the pathway. This radical pair (RP) initially forms with correlated spins that, as previous work on similar systems has indicated, could be influenced by an external MF. Based on a large body of subsequent work, which this review will not describe (for an in-depth review, see [3]), the generally accepted mechanism requires an RP, generated by photo-reduction of FAD, to undergo interconversion between the singlet and triplet states. The relative population of each spin state is altered by exposure to an MF. In the canonical model, the reverse reaction in CRY (electron returning to TrpH$^{\bullet+}$ from FAD$^{\bullet-}$) can only occur when the RP is in the singlet state. Thus, exposure to an MF is predicted to influence the probability of the reverse reaction occurring and thus modulate the half-life of "active" CRY, correlating with the flavin radical [4].

Whilst the CRY-RPM may provide an attractive explanation for a biological magnetoreceptor, until such a mechanism is shown to directly result in a physiological response, like an electrical response in a receptor cell, all mechanisms remain hypothetical. Thus far, an unequivocal demonstration has proven elusive. To enable biological testing of physical data requires model organisms that facilitate a combination of behavioural, cellular, molecular and genetic manipulation. The fruit fly, *Drosophila melanogaster*, offers these advantages, allowing either the entire nervous system or individual neurons to be genetically manipulated and subsequently tested for response to BL with/without an external MF by electrophysiology (Figure 1). Here, we summarise past and present research that shows this insect to be magnetosensitive through a CRY-dependent mechanism and highlight the studies that are uncovering the mechanistic basis for this "sixth sense".

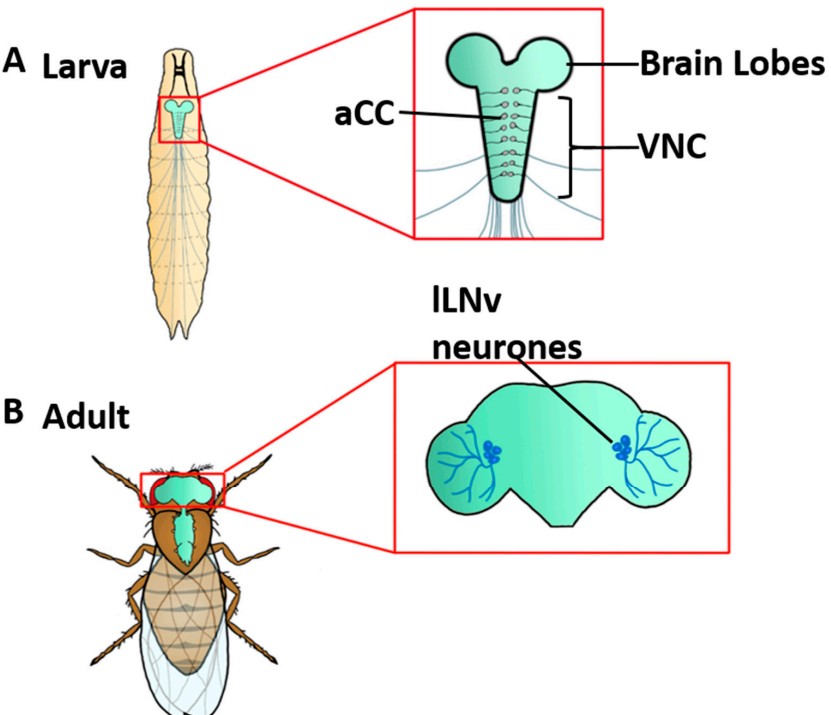

**Figure 1.** Diagrammatic representation of larval and adult *Drosophila*. (**A**) Representation of a *Drosophila* larva, with central nervous system (CNS) shown in blue. The expanded region shows the CNS with the approximate location of the segmentally repeated anterior corner cell (aCC) motoneuron along the dorsal midline of the ventral nerve cord (VNC). (**B**) Representation of adult *Drosophila* with CNS shown in blue. Expanded region shows the central brain region with approximate location of lLNv clock neurons.

Of course, an often-voiced criticism of using *Drosophila*, and indeed other insects, to study magnetosensitivity is that they do not navigate. This is certainly true, but it does not negate the fact that *Drosophila*, and other insects, have been unequivocally shown to be able to sense, and to be influenced by, applied MFs [5]. Indeed, all animals so far studied, from insects through to birds, seemingly share this ability, indicative perhaps that magnetosensitivity is a primitive sense. Equally, it seems probable that in animals (including the Monarch butterfly *Danaus plexippus*) that do navigate, this sense has been further refined to provide not only magnetosensitivity but also to act as a compass. Thus, regardless of this objection, *Drosophila* provides a highly tractable "living test-tube" to explore the mechanistic basis of magnetosensitivity in a biological system.

## 2. *Drosophila* Magnetosensitivity Requires the Presence of Cryptochrome

In *Drosophila*, numerous light-dependent roles have been described for CRY. Its best-described role is in circadian photoentrainment, but it also contributes to the regulation

of visual perception, UV avoidance and light-dependent arousal [6,7]. The first report of perception of the geomagnetic field by *Drosophila*, based on similar previous reports using other insects, was published in 1970 [8]. This observation was replicated, and extended, in a 1993 study that showed that *Drosophila* exhibit a wavelength-dependent response, indicating that they possess a light-dependent magnetic sense [9]. The first study to link magnetosensitivity to CRY was reported by collaborative research from the Reppert and Waddell groups [10]. These authors used a standard two-choice assay (a T-maze) to pair exposure to an MF (stated as 5G, G = geoMF) with a sucrose reward. Effects at 1G were not statistically significant. After successful training, adult *Drosophila* showed a clear preference for the arm of the T-maze exposed to the MF (now in the absence of the sucrose reward) relative to the arm under sham exposure. This preference was strongest under UV/BL (~420 nm), which is consistent with photoactivation of FAD, the cofactor of CRY. One inconsistent observation, however, was that preference was randomized under light >420 nm (where one might expect wavelengths up to 480 nm to be effective). This may, of course, be due to other confounding factors such as light power used. Notably, magnetosensitivity was lost in two independent loss-of-function mutations: $cry^{02}$ (a complete null) and $cry^b$ (loss-of-function) [10].

The Reppert group published two follow-up studies. In 2010, they exploited the relative ease of *Drosophila* genetics to show that a loss of magnetosensitivity in the $cry^b$ mutant was rescued by expression of DmCRY and by DpCRY1 and DpCRY2 (*Danaus plexippus* type I and II cryptochromes) [11]. Notably, mammals only express type II CRY (two variants, confusingly named CRY1 and CRY2) and it has been shown that HsCRY2 mostly restores magnetosensitivity in *Drosophila* $cry^b$ mutants [12].

Behavioural experiments are notoriously difficult to replicate, primarily because results can be (and usually are) influenced by a host of external influences ranging from ambient light levels and differing food ingredients used to raise animals through to genetic differences between presumed isogenic genotypes that have, nevertheless, been maintained in relative isolation for several years in differing laboratories. In addition to this, the field of magnetoreception brings its own environmental control issues, including the confounding effects of the geomagnetic field, erroneous electromagnetic noise, radio frequency interference and even heat and vibrations from the coils used to generate the exposure to MF. Thus, caution must always be used when interpreting data from single groups of researchers. In this regard, it is reassuring that additional groups, including the Ahmad/Helfrich-Förster, the Kyriacou/Rosato and the Chae groups report magnetosensitivity in *Drosophila* using independent behavioural assays.

The Ahmad/Helfrich-Förster group was the first to show that adult flies exposed to an external MF (static, 300 µT), under constant BL (445–495 nm), responded by lengthening their circadian period relative to control flies not exposed [13]. This effect was lost in $cry^b$ and $cry^{OUT}$ mutants. The Kyriacou/Rosato group reported a similar BL (430–470 nm) and MF-dependent effect on circadian period, but in this instance observed a shortening [14]. This effect occurred using field strengths as low as 90 µT, and in either static or oscillating MFs (3 or 50 Hz). The effect was dependent on the presence of CRY, being lost in $cry^{02}$ flies. That these two groups show opposite effects to circadian period is speculated to be due to an "antagonistic" effect of CRY due to the differing wavelengths used. Indeed, the Kyriacou/Rosato study reported that extending the wavelength to 500 nm (approximating the earlier study) resulted in a lengthening of circadian period [14]. Longer wavelengths may alter the triplet–singlet ratio and, therefore, affect the yield of active CRY [15].

The Kyriacou/Rosato group also reported a CRY- and MF-dependent locomotor phenotype of hyperactivity in adult flies. This same lab also demonstrated that adult flies exposed to both BL (450 nm) and an MF (500 µT) exhibited a reduced natural tendency for climbing (termed negative geotaxis) [16]. A similar study by the Chae group reported the same, yet independently produced, outcome [17]. These studies, which better address navigation through three-dimensional space, showed that exposure to an MF, similar to the geomagnetic field (71 µT for the Chae study), produced robust and significant

positive geotactic responses. Again, this behaviour was greatly reduced in the absence, or RNAi-mediated knockdown, of *cry* [16,17].

Other insects, in addition to *Drosophila*, have been used to study magnetosensitivity. For example, honeybees can be trained to associate a reward (sucrose) with exposure to an MF that resembled the geomagnetic field in strength (65 μT) [18]. The cockroach (*Blatella germanica*) has been shown not only to be magnetically sensitive but also to be able to orientate with respect to the direction of an applied MF, a key pre-requisite for navigation. Magnetosensitivity in this insect species was lost on silencing *BgcryII*, but not *BgcryI* (this cockroach, like the Monarch butterfly, expresses both types of CRY). Consistent with FAD photoreduction, a maximal response was observed with light of 365 nm (around the peak absorption that corresponds to the transition to the $S_2$ electronic excited state), which continually weakened up to 505 nm, after which the response rapidly diminished [19,20]. Whether this loss is due to the inability of light above ~500 nm to photoexcite FAD, or the antagonistic effect of short vs. longer wavelengths on CRY activation, a proposal based on RPM theory of magnetoreception [21,22], remains to be determined.

### 3. Mechanistic Basis of Magnetosensitivity in *Drosophila*

Type I CRY, as expressed in *Drosophila*, is photoactive due to bound FAD, which in darkness (i.e., inactive) is oxidised [23]. CRY-FAD has two broad absorbance peaks, 365 nm (UV) and 450 nm (blue) [24–26]. Biophysical assays, from several model systems (including *Drosophila* and the thale cress *Arabidopsis thaliana*), have investigated the photochemistry of CRY following BL/UV exposure to elucidate the signalling mechanisms at play. Accumulation of $FAD^{\bullet-}$, following exposure of AtCRY1 to BL, has been proposed to represent the "signalling state" of activated CRY [25,27]. The downstream signalling following this RP generation is still, however, unclear. Convincing experimental evidence suggests that photo-induced electron transfer triggers a conformational change in *Drosophila* CRY, which releases its C-terminal tail (CTT), opening binding sites for, as yet unidentified, downstream partners [4]. The role of DmCRY in the photo-entrainment of the fly circadian clock is thought to act through a similar mechanism (Figure 2). Under dark/unexposed conditions, the CTT acts as a repressor for DmCRY signalling, effectively blocking a binding pocket for downstream intermediates [28]. Upon exposure to light, this repression is released, promoting an interaction with the core clock protein Timeless (TIM), targeting CRY-TIM for proteasomal degradation [29,30]. The CTT contains motifs, including a region of protein–protein interaction and both PDZ- and calmodulin-binding domains [31,32]. Thus, release of the CTT, following light exposure, reveals a region that regulates downstream interactions that may bring about molecular, neuronal, and ultimately behavioural changes to the animal.

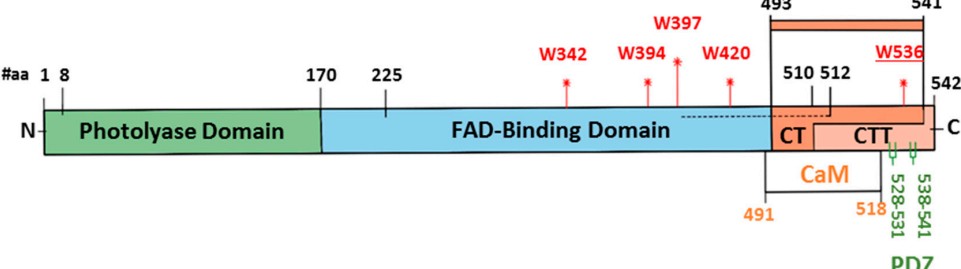

**Figure 2.** Representation of the key regions, motifs and residues in *Drosophila* CRY. Respective positions of structural elements/motifs are shown by amino acid number (#aa). Key regions include N-terminal Photolyase Domain (1-170aa), FAD-Binding Domain (171-512aa), and the C-terminus (CT) (493-541aa), which encompasses the important regulatory region of the C-terminus tail (CTT) (510-542aa). Tryptophan (W) residues implicated to support the canonical radical pair mechanism (RPM) are shown in red (although W536 is a residue of interest, it is yet to be investigated). Calmodulin-binding (CaM) (491-518aa) and PDZ (528-531 and 538-541) domains are present in the CTT. These are of particular interest as nucleation points for the recruitment of interactors and downstream signalling partners.

During the search for the elusive mechanism of CRY-dependent magnetoreception over the last decade, focus has been drawn to the generation of reactive oxygen species (ROS) during the flavin reoxidation step (i.e., the return to inactivated state) that follows CRY light exposure—an effect observed in both insect and plant CRY's. Exposure of AtCRY1 to BL results in protein activation and the subsequent production of ROS and hydrogen peroxide ($H_2O_2$) [33,34]. It is significant that the depletion of $H_2O_2$, via overexpression of the redox regulating protein catalase, is sufficient to block the DmCRY- and BL- dependent increase in action potential firing in the clock neurons of the fly (discussed below) [35]. Further investigation demonstrated that ROS are indeed produced, particularly during the relaxation of FAD back to its oxidized ground state [36], and that both ROS and $H_2O_2$ accumulate in the nucleus following BL exposure in plants [37]. Expression, of either AtCRY1 or DmCRY, in insect Sf21cells, also leads to the accumulation of ROS under light exposure, as well as transient superoxide radical formation ($O2^{\bullet-}$) [34,38]. The above studies provide provocative evidence for a redox-mediated signalling state, first hypothesised by the Holmes group [35]. This evidence is consistent with the proposal that, in the fly, CRY mediates cellular changes, such as to the excitability of neurons, via a change of cellular redox (e.g., Figure 3).

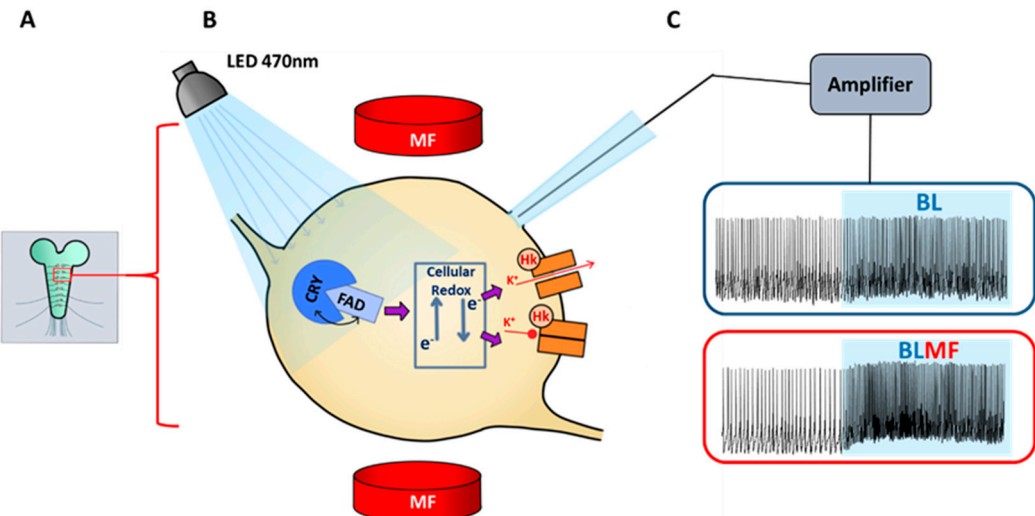

**Figure 3.** Diagrammatic representation of a patch-clamp electrophysiological recording from a *Drosophila* larval aCC motoneuron ectopically expressing DmCRY (**A**) An isolated larval central nervous system (CNS) is placed in a recording chamber between two static bar magnets (100 mT) with a patch electrode recording from an aCC motoneuron. (**B**) Cryptochrome is sufficient to impart BL sensitivity. The canonical mechanism suggests that illumination with blue light (BL) results in electron transfer between CRY and its cofactor FAD. In this active state, the conformation of CRY is altered, and the C-terminal of the protein, a region known to be important for interactions with other proteins, is exposed. How the activation of CRY signals to the neuron is not understood. However, the redox state of the cell has been shown to be an essential link between CRY activation and the depolarisation of the neuron through the closure of potassium channels, signalled via the intrinsically redox-sensitive Kvβ potassium channel subunit HYPERKINETIC (HK) [35]. (**C**) A closure of potassium channels leads to retention of positively charged potassium ions in the cell and a positive change in the membrane potential of the neuron. This can be observed by "patching the neuron" with an electrode. Following BL (light blue box), a rapid increase in both membrane potential and in action potential firing is observed. This effect is potentiated in the presence of a 100 mT magnetic field (MF).

## 4. CRY Mediates Magnetosensitivity in *Drosophila* Neurons

As we have seen, photoexcitation of CRY-FAD generates electron transfer and subsequent reduction of FAD, generating a [$FAD^{\bullet-}$ $TrpH^{\bullet+}$] RP in a spin-conserved manner. We also know that for CRY, from both *Arabidopsis* [39] and *Drosophila* [40] in the presence of an MF, the probability of the reverse reaction occurring in vitro is altered, which may in turn impact the population of "active" CRY [4]. For this to result in the magnetically

induced behavioural changes that have been observed [10], however, there must be a CRY- and MF-dependent effect on a receptor cell.

Direct recordings, using patch-clamp experiments, have shown that *Drosophila* adult clock neurons (Figure 1), which express DmCRY, respond to exposure to BL by increasing their action potential firing rate [41,42]. Significantly, ectopic expression of DmCRY in neurons that do not endogenously express this flavoprotein is also sufficient to confer the same response to BL [41,43]. In both scenarios, light-induced increases in neuronal excitability seemingly arise from the closure of voltage-gated potassium channels [41,43] (Figure 3). This alteration of neuronal activity requires the presence of HYPERKINETIC (HK) [35], a $Kv_\beta$ potassium channel subunit, which is itself redox-sensitive due to an intrinsic aldo-keto-reductase domain [44]. Thus, a favoured mechanistic understanding, at present, predicts that activation of DmCRY can increase neuronal activity via photo-induced changes to protein structure, allowing for the activation of downstream signalling mechanisms, and/or alteration of the cellular redox state, including influencing Kv channel activity [35]. The latter mechanism is particularly intriguing because an identical redox- and HK-dependent mechanism has recently been shown to modulate sleep in *Drosophila* [45]. Accumulation of ROS in cells of the *Drosophila* dorsal fan-shaped body, through mitochondrial activity, leads to the oxidation of HK-bound NADPH to NADP+. This, in turn, mediates the closure of $K^+$ channels and consequently increased neuron activity, correlating with an increased sleep pressure. HK couples with pore-forming $Kv_\alpha$ channels (in particular SHAKER and EAG) [35], thus mediating the transduction of cellular oxidative state to changes in $K^+$ flux and neuron excitability [35,41,44]. Although DmCRY is not seemingly required for sleep, it is perhaps indicative that the basic mechanism exploited by CRY may be evolutionarily old (being shared with sleep) and, as such, likely present in most neurons and across most, if not all, multicellular species.

Recent studies from our group report direct observations of an MF effect on neuron activity. Thus, we showed that exposure of *Drosophila* embryos to pulsed BL (470 nm) was sufficient to disturb neural development and to induce a seizure-phenotype in later larvae. This effect was greatly potentiated when repeated in the presence of a static MF (100 mT), but absent in *cry* nulls, or when orange light (590 nm) was used [46]. Use of a static 100 mT MF has a range of benefits over the μT field exposures immediately relevant to animal magnetoreception. First, the RP mechanism predicts that fields of this magnitude will saturate the Zeeman effect (splitting of different spin-states in terms of energy in the presence of a static MF) of typical organic RPs [47]. This is likely to produce a magnetically induced change in spin-selective product yield and reaction kinetics larger than those expected from μT fields and therefore might produce a larger physiological and organism response. Second, potential variations in background field are much less significant when using mT exposure conditions compared to μT conditions. Finally, the use of permanent magnets removes the confounding variables of vibration and heating that are possible when using the electromagnets necessary for μT exposure. In a follow-up study, we reported that coupling BL exposure with a static MF (100 mT) is sufficient to potentiate the effect of activated DmCRY on increasing the firing rate of action potentials in an identified motoneuron, termed the anterior corner cell, aCC (Figure 1). Again, no effects of either BL or MF were observed without prior expression of DmCRY or to orange light (590 nm) [43]. The effect was also abolished under conditions where Kv channels were blocked, suggestive of a similar, if not identical, mechanism to that proposed for clock cells requiring HK.

## 5. Is Full-Length CRY Essential for Magnetosensitivity in *Drosophila*?

In addition to showing that DmCRY is necessary for magentosensitive changes to circadian period, the study by Fedele et al. showed that the same effect was observed when only a fragment of this protein—the last 52 amino acids—was expressed [14] (Figure 2). This startling result suggests that magnetosensitivity may be supported by multiple mechanisms because this smaller protein fragment (termed CT, which contains and ends with

the C-terminal tail, CTT) lacks the FAD-binding domain and the TRP "chain" residues (W394, W342, W397, W420) critical to the canonical RPM [3,48–50]. Additional evidence also supports alternative mechanisms: these include observations that mutations to these same Trp residues, including W420F and W342F, at best attenuate but do not abolish the functionality of DmCRY [11,14,43,51]. Alternatives to the "canonical" RP between FAD$^o$– and TrpTH$^o$+, include the formation of an RP between FAD$^o$-/FADH$^o$ and O2$^o$– or another (unknown, Z$^o$) radical that intervenes in the light-independent oxidation of FAD$^o$-/FADH$^o$ to FAD and between FAD and other (non-Trp) amino acid residues in CRY. These "unconventional", RPs may contribute to magnetoreception, or may even be the actual magnetoreceptor [2,52,53].

The fact that the C-terminal alone is seemingly sufficient to transduce magnetosensitivity is intriguing. The C-terminal tail, known to be important in facilitating interactions between CRY and downstream partners, may possibly play a role [32]. The C-terminal does contain a sole Trp residue (W536), which is yet to be manipulated experimentally. It is, thus, possible that free FAD may be able to form an RP with this Trp residue, although it seems unlikely that FAD could bind directly to the C-terminal. Structural analysis of vertebrate Type II CRYs also indicates that they do not have a high affinity for FAD and are unlikely to bind significant quantities in vivo [54], even though they can support magnetosensitivity [12]. However, it seems clear that exposure to BL is a key factor in all assays that have measured magnetosensitivity. This raises the exciting possibility that free FAD (which is BL-sensitive) may form RPs with available Trp residues in other intracellular proteins. However, a necessity for CRY, or at least the C-terminal, to transduce this effect suggests that CRY is required for downstream signalling following RP formation, whether this is due to an RP formed with the Trp of the C-terminal or with those of other unidentified proteins. In this respect, it is possible that the previously discussed protein–protein interaction domain located in the CT fragment may provide a nucleation point for further components of this signal transduction cascade to assemble into a functioning magnetosensory complex.

## 6. Outlook

Significant evidence exists to implicate that CRY is a magnetoreceptor in animal cells. The majority of in vivo evidence is from experiments with *Drosophila*; indeed, there would be little direct experimental verification for the role of CRY in magnetoreception without this model organism. However, accumulating conflicting data may be telling us that multiple, perhaps even redundant, magnetosensitive mechanisms exist, which do not all require this flavoprotein. Whether CRY (or its C-terminal) is the primary magnetoreceptor, or indeed is an absolute requirement for magnetosensitivity, remains to be definitively shown. It is conceivable that CRY functions as an amplifier to boost weak MF effects that occur within free FAD. When not bound to protein, FAD can undergo intramolecular electron transfer between the adenine and the isoalloxazine chromophore following photoexcitation (i.e., auto-reduction) to form an RP. It was previously believed that this photochemistry was insensitive to external MFs at physiological pH. However, a recent study reports that FAD RPs are MF-sensitive at physiological pH, albeit to a lesser extent than at acidic pH [55]. Moreover, a new study has revealed that flavin autofluorescence in HeLa cells is sensitive to MF [56]. Although this is not a functional response, their observations are consistent with an underlying RPM and represent an exciting example of an MF effect on a chemical reaction in living cells. *Drosophila* represents the perfect test-bed to establish whether such effects on flavins in cells can form the basis of a functional response to MF. Thus, magnetosensitivity, through intramolecular RP formation within free FAD, may represent an ancient and ubiquitous signalling mechanism present in all cells, the efficacy of which is dependent on additional proteins, including CRY. To make progress in untangling conflicting data and to test novel hypotheses will require an ability to test potential mechanisms in biological tissue. The fruit fly, *Drosophila*, offers this potential through its ability to link genetics to physiology in identified neurons and to extrapolate findings to behavioural analyses in the presence of external MFs.

**Author Contributions:** All authors contributed to the writing of this article. All authors have read and agreed to the published version of the manuscript.

**Funding:** This research was funded by the Leverhulme Trust (RPG-2017-113, RAB and ARJ). Research in the group of RAB benefited from the Manchester Fly Facility, established through funds from the University of Manchester and the Wellcome Trust (087742/Z/08/Z). A.R.J. also acknowledges the National Measurement System of the Department for Business, Energy, and Industrial Strategy for funding.

**Institutional Review Board Statement:** Not applicable.

**Informed Consent Statement:** Not applicable.

**Data Availability Statement:** Not applicable.

**Acknowledgments:** We thank Ezio Rosato (Leicester) for constructive comments.

**Conflicts of Interest:** The authors declare no competing financial interests.

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
