# Peer review of "Exploiting the Fruitfly, Drosophila melanogaster, to Identify the Molecular Basis of Cryptochrome-Dependent Magnetosensitivity"

_quantumrep, doi:10.3390/quantum3010007_

Round 1

Reviewer 1 Report

The manuscript by Bradlaugh et al. reviews the current status of knowledge about the role of cryptochomes acting as potential magneto-sensors. The manuscript is concisely written. Although almost perfect, the authors may wish to address a few minor points.

General comment

It may be worth adding a Figure of DmCRY to illustrate, e.g., those domains referred to as interacting partners of intracellular binding partners.

Minor

l.34: introduce “RP“

l.54: skip “more”

l.66: skip “selected”

l.97-100: rephrase “Notably, mammals only express…and it has been shown that HsCRY2 mostly restores magnetosensitivty in Drosophila cry(b) mutants.

l.102: skip (and usually are)

l.116: introduce/explain cry(OUT)

l.119: rephrase “as predicted this…” to “The effect…”

l.134: change Dmcry to CRY

l.145-147: skip “the possibility of” and “(as described above)”

l.152: skip (

l.165: skip “identified”

l.186: change to “…change of the cellular redox state.”

l.197: change to “…understood. However,…”

l.214: change to “…patch clamp experiments, have…”

l.225: change to “…cellular redox state, including…”

l.262: skip “also”

l.265: skip “…the C-terminal…”

l.289: skip “…lone…”

l.298: change to “…without this model organism.”

l.300: skip “Indeed”

l.302: change to “…conceivable that CRY functions…”

l.314: change to “…behavioural analyses in the presence…”

Author Response

A new Figure 2 is now added to show this information.

All minor edits have been changed (using TRACK CHANGES).

Reviewer 2 Report

This is a topical and interesting review article that brings together a plthora of research that position the fruitfly Drosophila melanogaster as an eminent model to study Cry-dependent magnetosensitivity.

It must be emphasized that one needs to differentiate between photosensitivity and magnetosensitivity. A good recent article to refer and cite is:

Damulewicz Milena, Mazzotta Gabriella M. (2020) One Actor, Multiple Roles: The Performances of Cryptochrome in Drosophila.Frontiers in Physiology 11 10.3389/fphys.2020.00099

Another article on the non-circadian functions of Cry that the authors may wish to cite is:

Foley LE, Emery P. Drosophila Cryptochrome: Variations in Blue. Journal of Biological Rhythms. 2020;35(1):16-27. doi:10.1177/0748730419878290

The authors may also consider the new discovery regarding cellular autofluorescence and its magnetosensitivity:

Ikeya N, Woodward JR. Cellular autofluorescence is magnetic field sensitive. PNAS. 2021;118(3). doi:10.1073/pnas.2018043118

A mention to how such studies can be effectively replicated in Drosophila to gain insights into magnetosensivity would be desirable.

Author Response

We are grateful to the reviewer for these thoughtful suggestions, which we agree will help us to make important distinctions. We have cited these papers in the following passage before we begin to discuss the putative role of CRY in Drosophila magnetoreception:

“In Drosophila, numerous light-dependent roles have been described for CRY [J. Biol. Rhythms. 2020, 35(1), 16; Front. Physiol. 2020, 11, 99]. Its best described role is in circadian photoentrainment, but it also contributes to the regulation of visual perception, UV avoidance and light-dependent arousal.”

We very much agree that this publication is timely and will enable us to highlight further potential utility of Drosophila to explore flavin-based MF effects in living cells. We have therefore added the following passage to our Outlook section.

“Moreover, a new study has revealed that flavin autofluorescence in HeLa cells is sensitive to MF [PNAS 2021, 118 (3), e2018043118]. Although this is not a functional response, their observations are consistent with an underlying RPM and represent an exciting example of a MF effect on a chemical reaction in living cells. Drosophila represents the perfect test bed to establish whether such effects on flavins in cells can form the basis of a functional response to MF.”